# Marine Alga *Ulva fasciata*-Derived Molecules for the Potential Treatment of SARS-CoV-2: An *In Silico* Approach

**DOI:** 10.3390/md20090586

**Published:** 2022-09-19

**Authors:** Haresh S. Kalasariya, Nikunj B. Patel, Amel Gacem, Taghreed Alsufyani, Lisa M. Reece, Virendra Kumar Yadav, Nasser S. Awwad, Hala A. Ibrahium, Yongtae Ahn, Krishna Kumar Yadav, Byong-Hun Jeon

**Affiliations:** 1Centre for Natural Products Discovery, School of Pharmacy and Biomolecular Sciences, Liverpool John Moores University, Byrom Street, Liverpool L3 3AF, UK; 2Microbiology Department, Sankalchand Patel University, Visnagar 384315, India; 3Department of Physics, Faculty of Sciences, University 20 Août 1955, Skikda 21000, Algeria; 4Department of Chemistry, College of Science, Taif University, P.O. Box 11099, Taif 21944, Saudi Arabia; 5Reece Life Science Consulting Agency, 819 N Amburn Rd, Texas City, TX 77591, USA; 6Department of Biosciences, School of Liberal Arts & Sciences, Mody University of Science and Technology, Lakshmangarh, Sikar 332311, India; 7Department of Chemistry, King Khalid University, P.O. Box 9004, Abha 61413, Saudi Arabia; 8Biology Department, Faculty of Science, King Khalid University, P.O. Box 9004, Abha 61413, Saudi Arabia; 9Department of Semi Pilot Plant, Nuclear Materials Authority, El Maadi, P.O. Box 530, Cairo 11381, Egypt; 10Department of Earth Resources & Environmental Engineering, Hanyang University, 222-Wangsimni-ro, Seongdong-gu, Seoul 04763, Korea; 11Faculty of Science and Technology, Madhyanchal Professional University, Ratibad, Bhopal 462044, India

**Keywords:** SARS-CoV-2, *Ulva fasciata*, iGEMDOCK, PASS, *Swiss ADME*, VEGA QSAR, Schrödinger, Desmond software

## Abstract

SARS-CoV-2 is the causative agent of the COVID-19 pandemic. This in silico study aimed to elucidate therapeutic efficacies against SARS-CoV-2 of phyco-compounds from the seaweed, *Ulva fasciata*. Twelve phyco-compounds were isolated and toxicity was analyzed by VEGA QSAR. Five compounds were found to be nonmutagenic, noncarcinogenic and nontoxic. Moreover, antiviral activity was evaluated by PASS. Binding affinities of five of these therapeutic compounds were predicted to possess probable biological activity. Fifteen SARS-CoV-2 target proteins were analyzed by the AutoDock Vina program for molecular docking binding energy analysis and the 6Y84 protein was determined to possess optimal binding affinities. The Desmond program from Schrödinger’s suite was used to study high performance molecular dynamic simulation properties for 3,7,11,15-Tetramethyl-2-hexadecen-1-ol—6Y84 for better drug evaluation. The ligand with 6Y84 had stronger binding affinities (−5.9 kcal/mol) over two standard drugs, Chloroquine (−5.6 kcal/mol) and Interferon α-2b (−3.8 kcal/mol). *Swiss ADME* calculated physicochemical/lipophilicity/water solubility/pharmacokinetic properties for 3,7,11,15-Tetramethyl-2-hexadecen-1-ol, showing that this therapeutic agent may be effective against SARS-CoV-2.

## 1. Introduction

Marine macroalgae (also known as seaweed) are macroscopic, multicellular, eukaryotic photosynthetic organisms that belong to the Plantae kingdom [1]. These salt-dwelling marine plants are found on rock surfaces, corals, shells, pebbles, other plants, as well as the seabed or solid underlying layers of rock. Marine algae commonly grow in tidal and subtidal regions of the ocean in optimal availability of light. They can survive in harsh conditions (i.e., withstand heat, cold, UV radiation, salinity and desiccation) due to their easy adaptation to physiological changes by producing stress tolerant substances [2,3]. With this survival, they produce a variety of primary and secondary metabolites. Marine algae contain a plethora of biologically active compounds that have diversified therapeutic benefits [4,5]. They are taxonomically organized in three large and distinct groups, based on the color of the thallus and presence of pigments: brown algae (Phaeophyta), red algae (Rhodophyta) and green algae (Chlorophyta) [6]. These classes are better defined and more accepted than divisions. Seaweeds contain a high nutrient profile which includes polysaccharides, sterols, minerals, proteins, vitamins, fatty acids, lipids and carbohydrates that are being studied as potential therapeutic compounds [7,8]. With the presence of potential metabolites, phytochemical screening of marine algae has also revealed the presence of other compounds such as carrageenan, galactan, alginate, fucan, laminaran, sulfated polysaccharides, fucoidans, astaxanthin and others [9]. These substances possess antiviral activity and are currently gaining more consideration from pharmaceutical sectors for use in the development of an efficacious antiviral drug against the novel severe acute respiratory syndrome coronavirus 2 (SARS-CoV-2) [10].

The current pandemic coronavirus infectious disease 2019 (COVID-19) is caused by SARS-CoV-2, a zoonotic single-stranded RNA-enveloped virus that infects cells via its viral spike (S) protein [11]. As of 17 September 2022, the disease has infected 616,296,495 people and caused 6,526,820 deaths in 230 countries (according to Worldometer.info accessed on 1 September 2022). Genetic analysis found that the characteristics of SARS-CoV-2 exhibit different properties from previously known coronaviruses [12]. For example, the receptor-binding domain of this virus varies in several key amino acid residues that create a much stronger affinity for the transmembrane protein ACE2 (angiotensin-converting enzyme 2); however, there are several other proteins that may serve as coreceptors. After attachment, the virus utilizes receptors and endosomes of host cells to gain entry and once it enters the cell, viral proteins are assembled and synthesized that encode for the replicase-transcriptase complex [13]. These enzymes synthesize RNA via the viral RdRp (RNA-dependent RNA polymerase). Synthesis of structural proteins leads to the completion of assembly and the release of mature virus particles [14]. Conventional therapies may be insufficient to discourse the challenge of certain emerging and reemerging viruses and only a few antiviral therapeutic drugs may be helpful in controlling the spread of SARS-CoV-2 infection [15]. Ikegame et al. [16] suggested that vaccines might be ineffective in the cases of SARS-CoV-2 variants with mutations in the spike-encoding genes. Therefore, the design, discovery and development of broad-spectrum natural antiviral therapeutic agents that bind to particular SARS-CoV-2 targets are urgently required to fight the global pandemic. 

Considerations for designing therapeutic molecules include the inhibition of viral pathogenic factors and blocking specific host receptors [17]. For example, potential therapeutic compounds such as ACE 2 target to S-proteins (spike proteins) and NSPs (non-structural proteins) of SARS-CoV-2. Furthermore, ACE 2 has also been found to neutralize viral antibodies, while other antiviral peptides have been proven to be prospective therapeutic agents against SARS-CoV-2 [18]. NSPs and enzymes are also likely drug targets which prevent viral synthesis and replication by blocking structural proteins from binding to host cell receptors [19].

### 1.1. Therapeutic Approaches against COVID-19

Drug design strategies for SARS-CoV-2 focuses on two features: first by regulating the host immune defense and second by targeting the pathogenicity of the virus particles. The first method mainly works by obstructing or interfering with the signal transduction pathways in host cells which aid viral multiplication, whereas the second method targets SARS-CoV-2 by suppressing other viral activities such as inhibition of viral replication, RNA synthesis, enzyme activities and viral self-assembly [20]. The most definitive effective treatment being used thus far is the administration of remdesivir; however, it is not US FDA approved for COVID-19 treatment. Other drugs such as chloroquine (CQ), hydroxychloroquine (HCQ), azithromycin and ritonavir have been widely used as treatment for other diseases. There are current clinical trials to study the effectiveness of colchicine, glucocorticoids and dexamethasone against COVID-19; nonetheless, various adverse effects have been reported [21]. Some clinical studies have suggested that introduction of ARB (angiotensin receptor blockers) or ACI (ACE inhibitors) to SARS-CoV-2 infected individuals reduced the threat of inpatient death and protected infected hypertensive patients without extending the course of the disease. It is important to understand, however, that long-term clinical investigations are needed [22]. The exploration of natural sources for new safe and effective pharmaceutical broad spectrum antiviral candidates against SARS-CoV-2 is desirable. 

### 1.2. Antiviral Potential of Seaweed-Based Bioactive Compounds

Marine algae are one of the plentiful sources of diversified biologically active compounds that display antiviral activities and contain potential pharmacologically active constituents. For example, in 1958 polysaccharides from marine algae were found to be one of the potent sources of antiviral inhibition against mumps and the influenza B virus [23]. Pujol et al. [24] suggested that the antiviral strength of the algae-derived sulfated polysaccharides depends on the overall sulfate content, along with the positions of the sulfate groups, composition of sugars and the molecular mass. They assayed the antiviral action of different algal-derived polysaccharides against four serotype dengue viruses and found that antiviral effects were strongest during viral adsorption and internalization. Hans et al. [25] also reviewed the antiviral mechanisms of certain carrageenan polysaccharides from red algae. Their findings showed that human herpes virus type 1 (HHV-1) and poliovirus were inhibited by carrageenan. Algae-derived sulphated polysaccharides have been reported to show high binding affinity for the S protein of SARS-CoV-2 that prevents binding to the heparan sulfate co-receptor in host cells, thereby inhibiting viral infection [26]. In addition, porphyridium-derived exopolysaccharides such as sulfated-polysaccharides and carrageenan inhibit the internal entry or adhesion of SARS-CoV-2 to host cells, thereby reducing SARS-CoV-2 multiplication [27]. In other studies, Pereira and Critchley [28] found that the algae-derived polysaccharide, chitosan, effectively inhibits the viral interactions of HCoV-229E, -OC43, -NL63 and -HKU1 with ACE 2 proteins. They also suggested that the polysaccharides of seaweed can inhibit the life cycle of viruses by the inactivation of virions before virus maturation. Seaweed polysaccharides are also responsible for inhibition of virus internalization and uncoating, inhibition of replication and transcription, thereby improving host immune response or blocking viral infection. Zaporozhets and Besednova [29] suggested that there was a trans-/cis-cleavage inhibitory action by dieckol (a derivative of the phaeophyta, *Ecklonia cava*) on the SARS-CoV-2 3CLpro (chymotrypsin-like cysteine protease). Metabolically active constituents such as flavanones, alkaloids, phenolics and flavanols play an inhibitory role on 3CLpro, TMPRSS2 (transmembrane serine protease type 2) and ACE 2, implicit in the replication of SARS-CoV-2. Additionally, HCoV may be suppressed by sulphated polysaccharides such as fucoidan and sulphated rhamnan that display the inhibition of both the expression and actuation of human epidermal growth factor receptor [30]. 

In other studies, Kim et al. [31] found that 12.50 mg/mL of *Undaria pinnatifida* extract completely inactivated feline calicivirus (FCV) infection of host cells following a 1-h incubation. In addition, Mendes et al. [32] evaluated and demonstrated five out of six extracts of *U. fasciata* (four revealed 100% inhibition) possess virucidal activity on human metapneumovirus (HMPV) and the capacity to interact with extracellular viral particles thereby mediating the infection. Gomaa and Elshoubaky [33] found that extracts of Rhodophyta, *Acanthophora specifira* and Phaeophyta, *Hydroclathrus clathratus*, inhibited the propagation of HSV-1 and RVFV (rift valley fever virus) with minimal cytotoxicity to host cells. 

Furthermore, Elshabrawy [34] suggested algae-derived constituents can also suppress viral proteases. More interestingly, dieckol (isolated from *Ecklonia kurome*) targets 3CLpro, inhibiting the multiplication of SARS-CoV-2. Millet et al. [35] also suggested that the red algae-derived griffithsin (GRFT) targets antiviral activity on various viral spike glycoproteins of SARS-CoV and MERS-CoV (Middle East Respiratory Coronavirus), preventing viral entry. In addition, GRFT may be effectual against SARS-CoV-2 by inhibiting virus approach, reverse transcriptase, integrase and protease activities as suggested by Lee [36]. They also described how green algae-derived ulvans proved effective in the development of natural therapeutic agents against SARS-CoV-2 infection. 

Various in silico studies were conducted by different researchers on antiviral activities of *Arthrospira*-derived metabolites. as well as some other marine algae such as *Laurencia papillosa*, *Gracilaria corticata* and *Grateloupia filicina*-derived compounds, on different target proteins of SARS-CoV-2 [37]. Fucoidan and crude polysaccharides extracted from *Undaria pinnatifida* sporophyll, *Laminaria japonica*, *Hizikia fusiforme*, *Sargassum horneri*, *Codium fragile* and *Porphyra tenera* were screened for antiviral properties against SARS-CoV-2. Researchers determined that the majority of the tested extracts from marine algae exhibited antiviral activities at an IC_50_ of 12–289 μg/mL against SARS-CoV-2 pseudo-virus in HEK293/ACE 2 cells, except for *Porphyra tenera* (IC_50_ > 1000 μg/mL). Among selected seaweeds, crude polysaccharides isolated from *S. horneri* showed the strongest antiviral activity (IC_50_ of 12 μg/mL). These compounds can also be employed further for in vitro and in vivo evaluations [38]. 

Computer models provide researchers with valid and rapid alternatives to laboratory bench experiments in the drug discovery process, especially when dealing with numerous compounds. The algorithm needed to identify and define therapeutic agent properties (descriptors) involves two requisites: (1) algorithmic definitions, useful to and understood by software, and (2) identifying molecular features capable of elucidating predictable desired characteristics. After the descriptor features are defined, biological effects must correlate with calculable quantities relevant to those desirable properties [39]. Therefore, this present study aims to predict the physicochemical and pharmacokinetics properties of potential SARS-CoV-2 therapeutic compounds isolated from seaweed using in silico methods.

## 2. Results

### 2.1. GC-MS Characterization and PubChem^®^ Study

In our GC-MS characterization study, a total of twelve phyco-compounds were obtained from the methanolic extract of the seaweed, *Ulva fasciata.* These twelve compounds were identified based on retention time, % peak area, molecular formula and molecular weight. Unknown compounds were compared with the spectrum of the known components stored in the NIST library version 2005. In the PubChem study, the name of each compound used is a search input in PubChem database for retrieval of chemical information, as tabulated in Table 1.

### 2.2. VEGA QSAR Study for Mutagenicity/Carcinogenicity/Toxicity of Therapeutic Agents

VEGA QSAR calculations were obtained for six different models/analyses and subsequently used to predict whether a compound was either toxic or nontoxic. Results showed that five therapeutic compounds: azelaic acid, *n*-pentadecanoic acid, hexa-hydro-farnesyl acetone, palmitic acid (PA) and palmitic acid ethyl ester (PAEE), were determined to be either non-mutagenic, noncarcinogenic or nontoxic. Compounds with all three properties are desirable for a therapeutic agent (see Table 2).

### 2.3. PASS Predictions of Therapeutic Compounds for Select Viruses

In our VEGA QSAR predictions, the five non-mutagenic, noncarcinogenic and nontoxic compounds were used to predict antiviral activity against six known viruses as determined by PASS (http://way2drug.com/passonline/). This computer program independently calculates the estimated predictive activity spectra of compounds as probable biological activity (Pa) and probable biological inactivity (Pi), respectively, as mentioned earlier. The result of a prediction is presented as a list of Pa and Pi values, sorted in descending order of the difference as per: (Pa − Pi) > 0. Since Pa and Pi are probabilities, values range from 0–1.00 and only for activities where Pa > Pi are considered as possible test compounds. In drug compounds with Pa > 0.7, the probability of finding biological activity experimentally is high. If 0.5 < Pa < 0.7, the probability of elucidating biological activity at the bench is reduced and the therapeutic compound is probably dissimilar to any known therapeutic agent. If Pa < 0.5, it will be more difficult to obtain experimental results for biological activity, but the probability of elucidating a structurally new pharmaceutical agent is increased [40]. Out of the compounds tested, four showed significant Pa values (Pa > 0.5) with the exception of hexa-hydro-farnesyl acetone (whose data is not shown). All selected phyco-compounds (azelaic acid, pentadecanoic acid, palmitic acid and ethyl palmitate) expressed potential antiviral activity on infectious agents such as picornavirus, rhinovirus, poxvirus, adenovirus, cytomegalovirus and influenza whereas hexa-hydro-farnesyl acetone revealed a Pa value less than 0.5. The potential therapeutics had Pa values between 0.5 and 0.7 and thus may be considered structurally novel therapeutic compounds against SARS-CoV-2 (see Table 3). 

### 2.4. Docking Interaction Analysis of SARVS-CoV-2 Target Proteins by AutoDock Vina

The lowest binding energy reflects a strong/optimal binding strength between ligands and target proteins [41]. The catalytic site of the COVID-19 main protease known as 6Y84 [42,43] was found to bind 3,7,11,15-Tetramethyl-2-hexadecen-1-ol at −5.9 kcal/mol. Within immunogenic regions derived from the SARS-CoV N protein, an HLA-A*2402 restricted epitope N1 has been identified as protein 3I6L [44]. Our results showed that azelaic acid obtained the lowest binding energy (optimal affinity) with target protein 3I6L (−5.1 kcal/mol). Furthermore, the SARS-CoV-2 protein 6LU7 (i.e., main protease M^pro^) [44,45] in complex with an N3 protein inhibitor, revealed significant binding energy with 3,3,5-Trimethylhexahydro-azepine at −4.4 kcal/mol) (refer to Table 4 for other ligands and their interaction with different target protein in terms of binding energy). Based on these results, we decided to visualize the 3D structure of 3,7,11,15-Tetramethyl-2-hexadecen-1-ol. 

Three-dimensional description of ligand-protein interactions are depicted by Discovery Studio Visualizer that take advantage of binding modes/energies or poses [46]. Interaction of 3,7,11,15-Tetramethyl-2-hexadecen-1-ol with 6Y84 protein was rendered in 3D, as depicted in Figure 1a. Non-bond interaction between receptor and ligand is illustrated in Figure 1b whereas 2D representation of 3D ligand binding site is expressed in Figure 1c. Indirectly, a Ramachandran Plot was used to validate the modeled protein structure based on the Phi and Psi values that determined the quality of the protein structure. Good quality turns reflect efficient and accurate docking results. Figure 1d reveals the value of the Phi/Psi angles possible for an amino acid involved in ligand interaction. (Green: Inside; Pink: Overlap; Turquoise blue: Hardsphere) Figure 1e illustrates different receptor surfaces for ligand protein interaction complexes. Moreover, 3,7,11,15-Tetramethyl-2-hexadecen-1-ol conjugated to 6Y84 rendered the most stable binding poses with the different amino acids of the 6Y84 target protein. Referring to Table 5, it is evident that the most stable binding pose was the ligand 3,7,11,15-Tetramethyl-2-hexadecen-1-ol with 6Y84 targeted to TYR A:239, GLY A: 275, TYR A: 237, LEU A: 271 (Van der waals); THR A: 199, ARG A: 131 (conventional H-bond); and LEU A: 286, LEU A: 287, LEU A: 272 (alkyl) with a binding energy of −5.9 kcal/mol.

### 2.5. Comparison of Binding Energies of SARS-CoV-2 Target Proteins with Standard Drugs

3,7,11,15-Tetramethyl-2-hexadecen-1-ol yielded the optimal results for stable drug-ligand bonding. Therefore, the following experiment is based on this therapeutic agent and its interactions with selected SARS-CoV-2 target protein 6Y84. Furthermore, 3,7,11,15-Tetramethyl-2-hexadecen-1-ol was compared to other known potential therapeutic compounds. 

#### 2.5.1. Docking Interactions between 3,7,11,15-Tetramethyl-2-hexadecen-1-ol, HCQ, CQ, MPN, IFN α-2b and Remdesivir

In Figure 2, 3,7,11,15-Tetramethyl-2-hexadecen-1-ol showed effective binding energy (−5.9 kcal/mol) with 6Y84, better than the binding energy reported by two standard antiviral drugs CQ (−5.6 kcal/mol) and IFN α-2b (−3.8 kcal/mol), whereas HCQ, MPN and remdesivir show −6.1 kcal/mol, −7.3 kcal/mol and −7.1 kcal/mol binding energies, respectively. 

#### 2.5.2. Binding Energies of 3,7,11,15-Tetramethyl-2-hexadecen-1-ol and 5 Other Standard Antiviral Drugs with SARS-CoV-2 Target Proteins

In the overall assessment seen in Figure 3, comparison with CQ-6Y84 (−5.6 kcal/mol) and Interferon α-2b (−3.8 kcal/mol), (3,7,11,15-Tetramethyl-2-hexadecen-1-ol)-3I6L exhibits more effective binding energy (−5.9 kcal/mol). Besides, this ligand showed its binding affinity for different target proteins such as -1P9S (−5.7 kcal/mol), -2BX4 (−4.5 kcal/mol), 3I6L (−3.6 kcal/mol), -6LXT (−4.1 kcal/mol), -6VXX (−5 kcal/mol), -6VYB (−4.8 kcal/mol), -6M17 (−5 kcal/mol), -5RE4 (−3.8 kcal/mol), -6VSB (−5.2 kcal/mol), -6LU7 (−4.4 kcal/mol), -6M03 (−4.4 kcal/mol), -5R7Z (−4.1 kcal/mol), -5R81 (−4.2 kcal/mol) and -6YB7 (−4.9 kcal/mol).

Moreover, as included in Table 4, other optimal poses for standard antiviral drugs belong to HCQ-6YB7 (−7.8 kcal/mol), CQ-6Y84 (−7.6 kcal/mol), MPN-6Y84 (−9.2 kcal/mol), IFN α-2b-6VYB (−4.5 kcal/mol), IFN α-2b-6VXX (−4.5 kcal/mol) and remdesivir-6YB7 (−9.3 kcal/mol). By targeting selected target proteins, on 6Y84, 5366244 reported the highest binding energy (−5.9) than other ligand molecules. This value was higher than CQ-6Y84 (−5.6) and IFN α-2b-6Y84 (3.8). This ligand molecule also reported −5.7 kcal/mol binding energy with 1P9S protein that was higher than HCQ-1P9S (5.6), CQ-1P9S (5.6) and IFN α-2b-1P9S (4). Likewise, 5280435 reported high binding energy value (−5.2) with 6M03 target protein than found in CQ-6M03 (5.8), HCQ-6M03 (5.6) and IFN α-2b-6M03 (3.9). Moreover, 118239 showed −4.9 kcal/mol binding energy with 6LU7 protein which was found closer to the result obtained from IFN (−5 kcal/mol), HCQ (−5 kcal/mol) and CQ (−5.2 kcal/mol). In addition, 2266 revealed high binding energy with 3I6L (−5.1 kcal/mol) than IFN α-2b-3I6L (−3.7 kcal/mol) and equal to HCQ (−5.1 kcal/mol).

##### RMSD

The above plot shows the RMSD evolution of a protein (left *Y*-axis). All protein frames are first aligned on the reference frame backbone and then the RMSD is calculated based on the atom selection. In this plot, ‘Lig fit Prot’ shows the RMSD of a ligand when the protein-ligand complex is first aligned on the protein backbone of the reference and then the RMSD of the ligand heavy atoms is measured. Figure 4a illustrates the amino acid sequences of protein A chain whereas Figure 4b reveals the RMSD evolution of a selected protein and ligand.

##### RMSF

On this plot, peaks indicate areas of the protein that fluctuate the most during the simulation (Figure 4c). Typically, it was observed that the tails (*N*- and *C*-terminal) fluctuate more than any other part of the protein. Secondary structure elements like alpha helices and beta strands are usually more rigid than the unstructured part of the protein and thus fluctuate less than the loop regions.

Figure 4d reports SSE distribution by residue index throughout the protein structure. Figure 4e above summarizes the SSE composition for each trajectory frame over the course of the simulation and Figure 4f above monitors each residue and its SSE assignment over time.

Ligand RMSF shows the ligand’s fluctuations broken down by atom, corresponding to the 2D structure in the top panel of Figure 4g. The ligand RMSF provides insight on how ligand fragments interact with the protein and their entropic role in the binding event. In the bottom panel, the ‘Fit Ligand on Protein’ line shows the ligand fluctuations, with respect to the protein. The protein-ligand complex is first aligned on the protein backbone and then the ligand RMSF is measured on the ligand heavy atoms.

The protein–ligand interaction in the molecular dynamic simulation study is interpreted by studying its H-bond properties, hydrophobic interaction, ionic interaction and water bridges in Figure 4h.

H-bonding properties: The current geometric criteria for a protein-ligand H-bond is: the distance of 2.5 Å between the donor and acceptor atoms (D—H···A); a donor angle of 120° between the donor-hydrogen-acceptor atoms (D—H···A); and an acceptor angle of 90° between the hydrogen-acceptor-bonded atom atoms (H···A—X).

Hydrophobic interactions: The current geometric criteria for hydrophobic interactions are as follows: p-cation—aromatic and charged groups within 4.5 Å; p-p—two aromatic groups stacked face-to-face or face-to-edge; Other—a non-specific hydrophobic sidechain within 3.6 Å of a ligand’s aromatic or aliphatic carbons.

Ionic interactions: Ionic interactions or polar interactions are within 3.7 Å of each other and do not involve a hydrogen bond. We also monitor protein-metal-ligand interactions, which are defined by a metal ion coordinated within 3.4 Å of the protein’s and ligand’s heavy atoms (except carbon). 

Water Bridges: The current geometric criteria for a protein–water or water–ligand H-bond are: a distance of 2.8 Å between the donor and acceptor atoms (D—H···A); a donor angle of 110° between the donor-hydrogen-acceptor atoms (D—H···A); and an acceptor angle of 90° between the hydrogen-acceptor-bonded atom atoms (H···A—X).

##### Protein–Ligand Contacts

Figure 4i shows the total number of specific contacts the protein makes with the ligand over the course of the trajectory, whereas Figure 4j shows which residues interact with the ligand in each trajectory frame. Some residues make more than one specific contact with the ligand, which is represented by a darker shade of orange, according to the scale to the right of the plot.

Interactions that occur more than 10.0% of the simulation time in the selected trajectory (0.00 through 50.05 ns, are shown. 2D interaction of ligand atoms with the protein residues are depicted in Figure 4k. Figure 4l shows ligand torsion profile by a dial plot and bar plots. Each rotatable bond torsion is accompanied by a dial plot and bar plots of the same colour. Dial (or radial) plots describe the conformation of the torsion throughout the course of the simulation. The beginning of the simulation is in the centre of the radial plot and the time evolution is plotted radially outwards. The values of the potential are on the left *Y*-axis of the chart and are expressed in kcal/mol. Looking at the histogram and torsion potential relationships may give insights into the conformational strain the ligand undergoes to maintain a protein-bound conformation.

Ligand RMSD: The root mean square deviation of a ligand is shown with respect to the reference conformation (typically the first frame is used as the reference and it is regarded as time *t* = 0). Different ligand properties are illustrated in Figure 4m by separate plots. Radius of Gyration (rGyr) measures the ‘extendedness’ of a ligand and is equivalent to its principal moment of inertia. Intramolecular hydrogen bonds (intraHB) refer to the number of internal hydrogen bonds (HB) within a ligand molecule. Molecular surface area (MolSA) refers to the molecular surface calculation with 1.4 Å probe radius. This value is equivalent to a van der Waals surface area. Solvent accessible surface area (SASA) refers to the surface area of a molecule accessible by a water molecule. Polar surface area (PSA): solvent accessible surface area in a molecule contributed only by oxygen and nitrogen atoms.

Radius of gyration (RoG) is used to determine the compactness of a protein which is depicted in Figure 4n. When a protein is very compact, it tends not to fold easily. RoG is usually plotted after MD simulation for a protein–ligand complex. This elucidates the stability of the complex in addition to the RMSD.

### 2.6. Prediction of ADMET Properties for 3,7,11,15-Tetramethyl-2-hexadecen-1-ol

*Swiss ADME* calculated physicochemical properties as discussed in Section 4.10 and Section 4.11 for 3,7,11,15-Tetramethyl-2-hexadecen-1-ol: heavy and aromatic atoms (rings), FCsp^3^, rotatable bonds, H-bond acceptors/donors, MR and TPSA. The elucidation of these parameters in new therapeutic agents is paramount to undertaking clinical trials. Table 5 lists the physicochemical and lipophilic properties predicted by *Swiss ADME*.

#### 2.6.1. Heavy and Aromatic Atoms

The proportion of non-hydrocarbon atoms to non-hydrogen atoms defines the heavy atom proportion of a drug [47]. Heavy atoms were calculated by *Swiss ADME* to be 21 for 3,7,11,15-Tetramethyl-2-hexadecen-1-ol. Neither compound possessed any aromatic heavy atoms. Establishment of linear relationship between heavy atoms is one of the five molecular parameters and Log S (for Solubility) is useful in the water solubility prediction models. 

#### 2.6.2. Fraction Csp^3^

Fraction Csp^3^ has a positive correlation with successful drug development. If a compound can be rendered in three dimensions, then potential isomers of that compound may exhibit improved interactions with target proteins. Therefore, the drug specificity and potency are increased. The fraction of sp^3^ (FCsp^3^) hybridized carbons defines the levels of saturation and makes the proposed drug more three dimensional and therefore more effective. The highest FCsp^3^ values are likely to become therapeutic drugs and pass each stage of development [48,49]. Our results showed that 3,7,11,15-Tetramethyl-2-hexadecen-1-ol had a calculated Fraction Csp^3^ value of 0.90. Since this value is high, it is likely that this compound may be successful throughout the development process. 

#### 2.6.3. Rotatable Bonds

3,7,11,15-Tetramethyl-2-hexadecen-1-ol contained 13 rotatable bonds. The Rotatable bonds are helpful to know the molecular flexibility and to determine oral bioavailability of the compounds.

#### 2.6.4. H-Bond Acceptors (HBA) and Donors (HBD)

HBAs and HBDs are important parameters for both permeability and polarity of therapeutic agents. Compounds with more HBAs than HBDs are favored for drug agents. As stated earlier, Ro5 states that there should be ≤5 HBA and ≤10 HBD in a potential drug compound [49,50]. 3,7,11,15-Tetramethyl-2-hexadecen-1-ol had a total of 1 HBA and 1 HBD. This molecule satisfies the Ro5 conditions for hydrogen bonding.

#### 2.6.5. Molecular Refractivity (MR)

MR is a measure of the volume occupied by an atom(s) and calculated with the Lorenz-Lorentz equation [51]. MR reading was 98.94 for 3,7,11,15-Tetramethyl-2-hexadecen-1-ol. Molecular refractivity is one of the basic physicochemical properties to assess drug transport feature, mainly in regard to biological barrier crossings.

#### 2.6.6. Topological Polar Surface Area (TPSA)

Polar surface area is the sum of the contributions to molecular surface area of polar atoms (O, N and their attached H). TPSA is related to hydrogen bonding (O and N atom counts) and is important for permeability and oral bioavailability. TPSA uses the interactions of functional groups of a compound based on a database of chemical structures. If a TPSA value is <60 Å^2^ then that compound can be absorbed over 90%. Our study found that 3,7,11,15-Tetramethyl-2-hexadecen-1-ol possessed more optimal TPSA at 20.23 Å^2^ which is less than 60 Å^2^. This molecule can be readily absorbed into the GI tract. 

#### 2.6.7. Lipophilic Properties (Log P)

The lipophilicities of 3,7,11,15-Tetramethyl-2-hexadecen-1-ol were calculated by 6 mathematical models (software). 3,7,11,15-Tetramethyl-2-hexadecen-1-ol was found good in all the models (iLOGP (4.71), XLOGP3 (8.19), WLOGP (6.36), MLOGP (5.25) and Silicos-it (6.57). A consensus Log P was calculated by the arithmetic mean of the five models and determined to be 6.22. A compound with good lipophilicity can be considered a good candidate for a lipid base formulation, which has optimal physicochemical and ADME behavior.

### 2.7. Water Solubility, Pharmacokinetics, Drug Likeness and Medicinal Chemistry

Water solubility, GI absorption, skin permeability, P-gp, CYP inhibitions, Lipinski violations, bioavailability, PAINS alerts and synthetic accessibility are all required to analyze possible drug compounds (Table 6).

#### 2.7.1. Water Solubility by ESOL and Silicos-It Classes

The ESOL (estimated solubility) method of estimating aqueous solubility reported 3,7,11,15-Tetramethyl-2-hexadecen-1-ol to be moderately soluble (−5.98), whereas the Silicos-it chemoinformatic software calculated that this compound was moderately soluble (−5.51). Data reflects the GI absorption for this compound was low indicating that this therapeutic compound could be easily absorbed into the GI tract. Solubility is one of the beneficial parameters to understand the desired concentration of a compound for achieving essential pharmacological action. In qualitative estimation, the solubility class is given according to the following Log S scale: insoluble (more than −10), poorly (between −6 and −10), moderately (between −4 to −6), soluble (between −2 to −4), very soluble (between 0 and −2), highly soluble (more than 0).

#### 2.7.2. Pharmacokinetics

The GI absorption was found to be “Low” in 3,7,11,15-Tetramethyl-2-hexadecen-1-ol. This is a required step to assess drug behavior and optimal predelivery.

#### 2.7.3. Permeability Glycoprotein

P-gp (permeability glycoprotein) is the ATP-binding cassette transporter which plays a significant part in drug absorption and disposition, especially since it is found ubiquitously throughout the body including the intestinal lining. It also functions as a biological barrier by assisting in the removal of toxic compounds and xenobiotics outside of cells and limits cellular uptake of drug molecules or chemical entities from circulating blood into the brain and from intestinal organs [52]. Our results reflect that 3,7,11,15-Tetramethyl-2-hexadecen-1-ol functions as a P-gp substrate. Inhibition of P-gp can result in increased bioavailability of a drug and reduce the oral availability of drugs that are its substrate.

#### 2.7.4. Cytochrome P Inhibition

CYP-450 enzyme inhibitors increase the bioavailability and decrease the clearance, whereas non-inhibitors decrease the bioavailability and increase the clearance, of the drugs that cause adverse reactions. 

#### 2.7.5. Skin Permeability (Log Kp)

As stated previously, the more negative a molecule is for Log Kp, the less permeable it is. We found Log Kp to be negative in 3,7,11,15-Tetramethyl-2-hexadecen-1-ol (−2.29 cm/s).

### 2.8. Drug Likeness

Violations to Lipinski’s Ro5 are monitored at every step of drug synthesis or characterization of potential therapeutic agents. The bioavailability of a drug is also monitored throughout the developmental process. 

#### 2.8.1. Lipinski Violations

In the drug likeness category, 3,7,11,15-Tetramethyl-2-hexadecen-1-ol compound revealed 1 Lipinski violation. One violation means more than five H-bond donors, more than 10 H-bond acceptors, more than 500 Da molecular weight and log P over 5. The rule states that an orally active drug has no more than one violation.

#### 2.8.2. Bioavailability

Bioavailability measures how well a compound can be absorbed into cells of the body. Score predicts the probability of bioavailability > 10% in rat in *Swiss ADME*. Bioavailability score was calculated and found to be 0.55 for 3,7,11,15-Tetramethyl-2-hexadecen-1-ol.

### 2.9. Medicinal Chemistry

In medicinal chemistry, the PAINS (pan assay interference) method allows for the identification of potential problematic fragments, hitters or promiscuous compounds. Sometimes, these fragments yield false positive biological outputs. *Swiss ADME* provides an output of such faulty substructures. Nonetheless, when large datasets of chemical compounds prevent successful estimation by other means, PAINS estimations prove to be useful for pre-filtering large datasets by in silico approaches [53]. Our results show that the PAINS alert was zero for 3,7,11,15-Tetramethyl-2-hexadecen-1-ol. 

### 2.10. Synthetic Accessibility (SA)

In *Swiss ADME*, the SA demonstrated how this easy, rapid, helpful and robust a prediction method can be to aid in choosing potential therapeutic compounds to synthesize. The score ranges from 1 (very easy) to 10 (very difficult). The synthetic easy accessibility (SA) score was found to be 4.30 in 3,7,11,15-Tetramethyl-2-hexadecen-1-ol.

## 3. Discussion

There has been a considerable increase in evidence that reveals the antiviral action of various macroalgae derived metabolites, such as sulfated polysaccharides, phenolic compounds, fatty acid like compounds, pigments and other bioactive moieties [54,55,56,57,58]. This present study reported an effective binding of *U. fasciata* derived compound, 3,7,11,15-Tetramethyl-2-hexadecen-1-ol, with 6Y84 target protein of SARS-CoV-2 that represents the catalytic site of the COVID-19 main protease [59,60]. This binding revealed effective binding energy (−5.9 kcal/mol), better than the binding energy reported by two standard antiviral drugs CQ (−5.6 kcal/mol) and IFN α-2b (−3.8 kcal/mol).

The literature and data repositories were searched for previous studies of active ingredients of seaweed for interaction between the ligands and SARS-CoV-2 proteins. Tafreshi et al. [61] demonstrated that 6,6′-Bieckol (source: brown alga *Ecklonia cava*), Pseudotheonamide C (source: green alga *Sargassum spinuligerum*), 8,8′-Bieckol (source: brown algae *Ecklonia cava* and *Ecklonia kurome*) and Dieckol (source: brown alga *Ecklonia cava*) compounds are better in total interaction energy than standard antiviral compounds Remdesivir and Lopinavir against target to protein main protease (Mpro) in complex with an inhibitor N3 of SARS-CoV-2 (PDB ID:6LU7). Among 27 different algal derived molecules, sulphated tri-, tetra- and penta-saccharides from *Porphyridium* sp. exopolysaccharides (SEP) showed high affinities to SARS-CoV-2 Mpro protease co-crystallized with the inhibitor PF-07321332 (PDB ID: 7VH8). In a study carried out by Hlima et al. [62], three sulphated saccharides (CMA1, CMA3 and CMA5) revealed higher binding energy than Paxlovid. Among these molecules, sulphated penta-saccharide namely CMA3 has shown the best binding affinity toward Mpro with a binding energy of −9.9 kcal/mol. Muteeb et al. [61] performed virtual screening of 1110 seaweed derived ligands against 3CLpro of SARS-CoV-2. Based on docking score values, they selected nine compounds and further evaluated for finding effective therapeutic compound. They showed that callophysin A (source: red alga *Callophycus oppositifolius*) interacted with the catalytic residues (His41 and Cys145) of 3CLpro by showing mechanism-based competitive inhibition. Tassakka et al. [63] studied the therapeutic efficacy of compounds from marine red alga *Halymenia durvillei* against SARS-CoV-2 inhibition. They found that 1–2 tetra-decandiol and E,E,Z-1,3,12-nonadecatriene-5,14-diol were identified based on pharmacophore study, while cholest-5-En-3-Ol (3.Beta.)- had a high fitness score in molecular docking studies, both in monomer and dimer state, compared to the N3 inhibitor and remdesivir affinity scores against the 3CL-Mpro. Bharathi et al. [50] also reported that seaweed derived compounds including caffeic acid hexoside (−6.4 kcal/mol) and phloretin (−6.3 kcal/mol) from brown alga *Sargassum wightii* showed the inhibitory action against the crucial residues ASN417, SER496, TYR501 and HIS505, which are supported for the inviolable omicron and angiotensin-converting enzyme II (ACE2) receptor interaction. Likewise, Kumar et al. [56] reported 14 marine derived compounds with better docking scores than the reference compounds and with considerable molecular interaction with the active site residues of SARS-CoV-2 virus targeted proteins, 3CLpro, PLpro and RdRp. Additionally, Lira et al. [64] and Park et al. [65] showed that phloro-tannins isolated from marine algae *E. cava* were able to target SARS 3CLpro through in vitro studies. Besides, in silico molecular docking study reported the antiviral potential of different marine algae derived compounds, including oleic acid, saringosterol, b-Sitosterol, Caulerpin, glycoglycerolipids, Kjellmanianone and Loliolide, as potential inhibitors of target protein 3CLpro, the spike protein and the ACE-2 receptor of SARS-CoV-2 [66]. Similarly, Ray et al. [67] reviewed the potential role of red algae derived bioactive polysaccharides such as fucan sulfates, ulvan, alginate, agarans, carrageenans and galactans against different target proteins of SARS-CoV-2. They also reported the role of phloro-tannins derived from brown alga *Sargassum spinuligerum* for COVID-19 therapeutics. As reported in this literature, the therapeutic actions of seaweed-derived bioactive compounds may be applicable as a novel compound to combat infection of SARS-CoV-2. As noted, there is not any previous record of virtual screening for green alga *Ulva fasciata* against target proteins of SARS-CoV-2. Likewise, in seaweed species other than *U. fasciata*, researchers mainly target natural compounds to different receptors including 3CL-Mpro, PLpro and RdRp and 6LU7, but there is no previous record against 6Y84 receptor characterized for SARS-CoV-2 main protease with an unliganded active site (2019-nCoV, coronavirus disease 2019, COVID-19). According to Protein Data Bank Japan (PDBj), this protein plays its major role in peptidase activity and viral protein processing. In the present investigation, a total of five compounds out of twelve were found to be nontoxic, non-mutagenic and noncarcinogenic via the VEGA QSAR toxicity predictions tool. These potential therapeutic compounds targeted 15 viral proteins of SARS-CoV-2 where binding interactions by molecular docking were studied. In binding interaction studies, the 3,7,11,15-Tetramethyl-2-hexadecen-1-ol molecule revealed more negative binding energy as well as interaction with the target protein 6Y84 (−5.9 kcal/mol). 

Chemically, 3,7,11,15-Tetramethyl-2-hexadecen-1-ol is (E)-3,7,11,15-tetramethylhexadec-2-en-1-ol. It is similarly known as Phytol or 3,7,11,15-teramethyl-2-hexadecene-1-ol, (2E, 7R, 11R), which belongs to the fatty alcohol category according to MeSH classification. Phytol is a diterpenoid that is hexadec-2-en-1-ol substituted by methyl groups at positions 3, 7, 11 and 15. It is a diterpenoid and a long-chain primary fatty alcohol. Previous studies reported easy extraction and characterization of phytol from different marine algae species. It was isolated from the brown alga *Dictyopteris membranaceae* by supercritical fluid extraction (SFE) at 91 bar/40 °C, at 91 bar/40 °C, 1.8 g/min of CO_2_ flow rate and 35 min of extraction time (5 min static time + 30 min dynamic time) [68]. Likewise, Pejin et al. [69] and Hattab et al. [70] extracted phytol from edible marine algae by microwave assisted extraction, high-speed counter current chromatography and supercritical fluid extraction. (91 bar/40 °C, 1.8 g/min of CO_2_ flow rate and 35 min of extraction time) Xiao et al. [71] used microwave-assisted extraction coupled with high-speed counter-current chromatography of *Undaria pinnatifida* and *Sargassum fusiforme* for separation of phytol. They separated and purified 3.5 mg of the phytol obtained from 15.0 g *S. fusiforme,* and 10.7 mg phytol were obtained from 15.0 g *U. pinnatifida* with a non-aqueous two-phase solvent system composed of n-hexane–acetonitrile–methanol (5:5:3, *v*/*v*/*v*). Hence, this compound may become easy and economical to isolate, extract and characterize for use in therapeutic preparations due to its presence in different marine algal species and medicinal plants.

Using the Schrodinger Desmond package, a molecular dynamic simulation analysis was conducted for fifty nanoseconds to test the stability and flexibility of the protein–ligand complex. The finding showed that the protein–ligand complex was stable during the simulation. In molecular dynamic simulation study, non-specific interaction was found between 3,7,11,15-Tetramethyl-2-hexadecen-1-ol and its receptor, 6Y84. In this study, a non-specific hydrophobic sidechain within 3.6 Å of a ligand’s aromatic or aliphatic carbons was found (Figure 4h). Through MD simulation, ligand non-bond monitor was observed with the following patterns: N:UNK0:H—A:THR199:OG1; A:ARG131:HH21—N:UNK0:O; N:UNK0:O—A:ASP289:OD2; N:UNK0—A:LEU272; N:UNK0—A:LEU287; N:UNK0:C—A:LEU286; N:UNK0—A:LEU286; N:UNK0:C—A:LEU286.(UNK = 3,7,11,15-Tetramethyl-2-hexadecen-1-ol). This Ligand Non-Bond Monitor displays an interaction between pairs of atoms between the current ligand and receptor. Moreover, physicochemical, pharmacokinetic, drug-likeness and related parameters of both palmitic compounds were found to be useful in *Swiss ADME* predictions for future in vitro and in vivo experimentation [72]. As for the non-specific binding limitations, we believe that only solid in vitro and in vivo research can determine if they would cause any deleterious effects. Ultimately, our findings may lead to the use of 3,7,11,15-Tetramethyl-2-hexadecen-1-ol in disease treatment for SARS-CoV-2. This compound can be administered alone or as an adjunct, after further pharmacological evaluations at the bench, for future drug development [73,74]. Although, many seaweed derived molecules have shown promising antiviral actions in vitro and in vivo models, an interdisciplinary effort between academicians, clinicians and the industries would be required to fully characterize chemical compounds and their biological activity profiles and mechanism of antiviral action to progress the development of this molecule to address global unmet clinical needs [75,76]. Hence, considering the dire need for the development of therapeutics against SARS-CoV-2, there is a necessity to screen a myriad of macroalgae-derived potential antiviral compounds, which demands further evaluation and research. Many COVID-19 mutant strains have spread throughout the world and we can identify specific target proteins from the infectious strains and then target those proteins. However, detailed in vitro and in vivo studies are required to confirm the antiviral activity against this novel coronavirus. 

## 4. Materials and Methods

### 4.1. Site Location and Sample Collection 

Geographically, India has about a 7500 km coastline, which contains a highly diversified marine ecosystem where seaweeds are widely distributed. Seaweed is disseminated in India in the coastal region of Tamilnadu, Gulf of Kutch in Gujarat, Lakshadweep, Vishakhapatnam, Mandapam, Madras, Mahabalipuram and the coastal areas of Maharashtra and Goa [77,78]. Gujarat is one of the richest states in India for biodiversity of seaweeds and it covers 1600 km of coastline. The Gulf of Kutch represents a wide variation of seaweed diversity in different regions [79]. It is for this biodiversity that Gujarat was chosen as the location to obtain seaweed samples for the current investigations. 

A sample of marine macroalga, *U. fasciata*, was collected from Beyt Dwarka, Dist.-Devbhoomi Dwarka, Gujarat, India via a standard handpicking method [80] in March 2020. The location of the collection site was 22°28′47.9″ N, 69°08′05.0″ E (refer to Figure 5). An in situ image of *U. fasciata* in sea water as well as an image of an isolated sample is depicted in Figure 6.

### 4.2. Sample Preparation and Identification 

Before identification, the collected sample was cleaned with distilled water to separate out impurities, salts, particles, epiphytes and debris. Then the sample of *U. fasciata* was freeze-dried overnight (freeze dryer, Esquire Biotech, Chennai, India) before storing at −20 °C to preserve its morphological structures and biochemical composition. Sample identification [81] was performed with the help of Dr. N.H. Joshi, Aquaculture Department, Junagadh Agriculture University, Veraval, India.

### 4.3. Extract Preparation and GC-MS Characterization 

The sample was dried at 60 °C (REMI RDHO 80 Dry Hot Air Oven, Remi Elektrotechnik Limited, Goregaon, India), ground and preserved in desiccated plastic sterile bottles. The extract was prepared by adding 2 g of powdered algae sample with 20 mL methanol (anhydrous, 99.8%, Sigma-Aldrich, Bangalore, India). The mixture was then incubated overnight in an orbital shaker (REMI CIS 24 Orbital Shaker, Remi Elektrotechnik Limited, Goregaon, India) at 32 °C. The filtration of the extract was performed by using a Whatman No. 1 filter paper (Qtech Scientific India Private Limited, Faridabad, India) and evaporated under decreased pressure (150 mbar) at 20 °C using a rotary vacuum evaporator (Sigma Scientific, Nangainallur, Chennai, India). The concentrated extract was utilized for GC-MS analysis to identify resident phyco-compounds. 

GC-MS is a hybrid analytical technique that is helpful for the chemical characterization of compounds, which couples GC (gas chromatography) for the separation of chemical compounds with the detection properties of MS (mass spectrometry) to provide mass determination [82]. Phyco-compound analysis was done at the Sophisticated Analytical Instrument Facility (SAIF), IIT-Bombay, India. The GC-MS system (New Delhi, India) was a JEOL, AccuTOF GCV time-of-flight mass spectrometer. The specifications of this instrument and analysis were as follows: Agilent 7890 GC; Flame Ionization Detector; head space injector; combipal autosampler; EI/CI ionization modes; time-of-flight (TOF) analyzer; Ion Source temperature and Interface temperature: 250 °C; mass range: 50–600 mass units; flow rate of helium (He): 1 mL/min, ionization voltage: 70 eV; injection of sample in split mode as 1:10. 

Unknown phyco-compounds were identified by comparing with the standard data in the NIST library version 2005. The obtained compounds were used to further study biological activity and toxicity prediction as well as molecular docking interactions via in silico analysis. 

### 4.4. PubChem^®^ Study

PubChem^®^ is an open chemical structure database sponsored by the National Center for Biotechnology Information (NCBI), National Library of Medicine (NLM), National Institutes of Health (NIH), Department of Health and Human Services (DHHS), Bethesda, MD, USA (https://pubchem.ncbi.nlm.nih.gov/ accessed on 21 June 2022). It is a huge collection of information related to chemical nomenclature, chemical structures, identifiers, physical, chemical and biological properties, patents, health, safety, toxicity data and other descriptors. The PubChem^®^ database information is helpful in drug discovery by the utility of multiple programmatic access routes to complete virtual automated screening of chemical compounds. In addition, this database permits users to download PubChem^®^ data files in multiple formats and upload them into local computing facilities, enabling data integration between PubChem^®^ and other resources such as online browsing tools [83]. Hence, the retrieved information is useful for predicting biological effects, toxicity prediction, multitarget ligands and targets of constituents for drug-repurposing or off-target concomitant prediction. For this work, PubChem^®^ ID, molecular formula, molecular weight, CAS (Chemical Abstracts Service) no., EC (European Community) no. and canonical SMILE (Simplified Molecular-Input Line-Entry) structures were collected from this database and listed in Table 1. 

### 4.5. VEGA QSAR Toxicity Prediction Study

VEGA is a read across tool to extract values for target compounds on the basis of structurally related substances. By using the VEGA web platform, a series of QSAR (quantitative structure-activity relationship) models can be utilized to develop chemical models for regulatory/research purposes [84]. It is the ideal program for analyzing large datasets by modifying any of the data file formats into an internal compatible format. It can be easily installed and used on any operating system supporting JAVA. In addition, users usually prefer to employ this program due to its free availability and accessibility. VEGA includes the three main algorithms of a QSAR model that include toxicological, biological and chemical data used for structural modelling of each new compound. Moreover, this platform offers a facility that combines the predictive capability of computer-based models, with explanatory tools and toxicological endpoints, which may be convincing and useful for researchers in drawing a conclusion from the query. The resulting system covers a large set of toxicological endpoints. Using this platform, users can access a series of data from the VEGA program as a series of QSAR models after selection of SMILE or chemical structures as inputs [85].

Prediction was done by selecting six models of QSAR for different properties such as Mutagenicity, Carcinogenicity and Toxicity to select the best model as well as best compound with potent activities by refining each phyco-compounds. In this study, six QSAR models, the Mutagenicity (Ames test) CONSENSUS Model 1.0.3, the mutagenicity (Ames test) model 2.1.13, the carcinogenicity model (CAESAR) 2.1.9, the carcinogenicity oral classification model (IRFMN) 1.0.0, the developmental toxicity model (CAESAR) 2.1.7 and the Developmental/Reproductive Toxicity Library (PG) 1.1.0, were used to screen virtually all compounds with significant or poor medicinal properties for SARS-CoV-2 antiviral drug designing/development *in silico*.

### 4.6. PASS Predictions

Bioactive compounds may possess therapeutic or adverse effects on the body after administration. Properties making up the biological activity spectrum (BAS) of a compound’s chemical structure include mutagenicity, carcinogenicity, teratogenicity and embryotoxicity [86]. In our study, the BAS of each sample compound was evaluated by the PASS (prediction of activity spectra for substances) web-based application. For prediction, PASS requires an .sdf SMILE structure file or a .mol file as an input into the program. PASS can predict more than 1000 biological and toxicological activities from structural formulas to predict all effective phyco-compounds of potential therapeutic compounds. Predictions are compared to the available information on the pharmacological and toxicological characteristics of the selected compounds stored in the database (i.e., training sets) [87]. Hence, the PASS server becomes useful for predicting antiviral activity on other known viruses to elucidate possible compounds considered as antiviral agents against this novel SARS virus. In this study, SMILE structures of each potential therapeutic compound were used to achieve antiviral predictions against known viruses. This online server predicts over 4000 types of different biological activities but still does not contain any information for antiviral actions against SARS-CoV-2, since this virus and associated research is too new. 

### 4.7. SARS-CoV-2 Target Protein Selection 

The AutoDock Vina program was used to conduct a molecular docking study between selected phyco-compounds and target proteins on the SARS-CoV-2 virus [88,89] for the recognition of potential phyco-compounds and predictions of ligand–protein interactions. This docking tool is comprised of four steps in the virtual screening of a compound: (1) obtaining chemical structure of analyzed therapeutic agent, (2) calculating the compound library, (3) generation of protein-ligand docking and (4) docked poses for analysis [56]. For AutoDock Vina program, 15 target proteins were selected for study from the Protein Data Bank (PDB): 1P9S, 2BX4, 3I6L, 6LXT, 6VXX, 6VYB, 6M17, 5RE4, 6VSB, 6LU7, 6M03, 5R7Z, 5R81, 6YB7 and 6Y84. The .pdb files listing the target proteins were utilized as an input for further docking interaction predictions with various ligands. Target proteins and their structural characteristics and functions are listed in Table 7.

### 4.8. Selection of Standard Drugs and Docking Interaction Analysis

Docking interaction analysis is an important technique for drug discovery and to predict the binding affinity between a ligand molecule (.sdf file) and target proteins (.pdb file). Retrieved .sdf file of each phyco-compounds from PubChem database converted into .pdb file using translator tool (https://cactus.nci.nih.gov/translate/ accessed on 28 June 2022). These .pdb files were used as an input in AutoDock Vina program. This evaluation predicts the optimal orientations of ligand-to-protein binding affinities (i.e., poses) to predict the formation of a stable complex. Docking interactions were accomplished by an open-source program AutoDock Vina [99,100]. Five different standard drugs from the literature were selected to study binding interactions with specific target proteins: HCQ, CQ, methylprednisolone (MPN), IFN α-2b and remdesivir. These drugs were selected due to their effective inhibitory roles against SARS-CoV-2. Likewise, the mechanistic in vivo introduction of HCQ for the inhibition of COVID-19 is described by Ou et al. [101]. In clinical studies, CQ appeared to exhibit major potency against SARS-CoV-2 in COVID-19-related pneumonia cases [102]. Clinical trials found the use of MPN (also known as dexamethasone) proved to be better therapy of hospitalized COVID-19 patients [103]. Polyethylene glycol (PEG) IFN α-2b reduces the duration of SARS-CoV-2 viral shedding as per Phase 2 clinical trial data [104]. Lastly, remdesivir, a nucleoside analog, is the only US FDA accredited drug for the treatment of SARS-CoV-2 by inhibition of RdRp [105]. Binding energy values expressed by each potential phyco-compound were evaluated for efficacy on various viral targets. Phyco-chemical structural details such as molecular formulae, molecular weight, SMILE structures and the PubChem ID retrieved from the PubChem database, are listed in Table 8. Optimal ligands were next analyzed for pharmacokinetic properties along with molecular dynamic simulation analysis of ligand protein complex.

### 4.9. Molecular Dynamic Simulation Study

A molecular dynamic simulation study predicts how every atom in a protein or other molecular system will move over time, based on a general model of the physics governing inter-atomic interactions. It can capture a wide variety of important biomolecular processes, including conformational change, ligand binding and protein folding, revealing the position of all the atoms at femtosecond temporal resolution. Importantly, this simulation study can also predict how biological molecules will respond, at an atomic level, to perturbations [106,107]. Desmond from Schrödinger’s suite was used to predict molecular dynamic properties for ligand–protein interaction analysis. Schrödinger’s Suite is a software package which is used for lead discovery and lead optimization in the context of drug discovery, atomic scale simulation of chemical substances, modelling biomolecules and molecular graphics for communicating structural results. Desmond is integrated with molecular modeling environment (Maestro, developed by Schrödinger, Inc.) for setting up simulations of biological and chemical systems and is compatible with Visual Molecular Dynamics (VMD) for trajectory viewing and analysis [108,109]. This can be access from https://www.deshawresearch.com/downloads/download_desmond.cgi/ (accessed on 5 July 2022).

#### 4.9.1. RMSD and RMSF Calculation

The Root Mean Square Deviation (RMSD) is used to measure the average change in displacement of a selection of atoms for a particular frame with respect to a reference frame. It is calculated for all frames in the trajectory [110]. The Root Mean Square Fluctuation (RMSF) is useful for characterizing local changes along the protein chain [111].

Protein RMSD: Monitoring the RMSD of the protein can give insights into its structural conformation throughout the simulation. RMSD analysis can indicate if the simulation has equilibrated, i.e., its fluctuations towards the end of the simulation are around some thermal average structure. Changes of the order of 1–3 Å are perfectly acceptable for small, globular proteins. Changes much larger than that, however, indicate that the protein is undergoing a large conformational change during the simulation. It is also important that simulation converges—the RMSD values stabilize around a fixed value. If the RMSD of the protein is still increasing or decreasing on average at the end of the simulation, then the system has not equilibrated and the simulation may not be long enough for rigorous analysis.

Ligand RMSD: Ligand RMSD (right Y-axis) indicates how stable the ligand is with respect to the protein and its binding pocket. If the values observed are significantly larger than the RMSD of the protein, then it is likely that the ligand has diffused away from its initial binding site. 

#### 4.9.2. SSE and L-RMSF Determination

Protein secondary structure elements (SSE) such as alpha-helices and beta-strands are monitored throughout the simulation.

The Ligand Root Mean Square Fluctuation (L-RMSF) is useful for characterizing changes in the ligand atom positions.

#### 4.9.3. Protein-Ligand Contacts

Protein interactions with the ligand can be monitored throughout the simulation. These interactions can be categorized by type and summarized, as shown in the plot above. Protein–ligand interactions (or ‘contacts’) are categorized into four types: hydrogen bonds, hydrophobic, ionic and water bridges. Each interaction type contains more specific subtypes, which can be explored through the ‘Simulation Interactions Diagram’ panel. The stacked bar charts are normalized over the course of the trajectory: for example, a value of 0.7 suggests that 70% of the simulation time the specific interaction is maintained. Values over 1.0 are possible, as some protein residue may make multiple contacts of the same subtype with the ligand.

Hydrogen Bonds: (H-bonds) play a significant role in ligand binding. Consideration of hydrogen-bonding properties in drug design is important because of their strong influence on drug specificity, metabolization and adsorption. Hydrogen bonds between a protein and a ligand can be further broken down into four subtypes: backbone acceptor; backbone donor; side-chain acceptor; side-chain donor.

Hydrophobic contacts fall into three subtypes: p-cation, p-p and other non-specific interactions. Generally, these types of interactions involve a hydrophobic amino acid and an aromatic or aliphatic group on the ligand, but we have extended this category to include p-cation interactions.

Ionic interactions: or polar interactions, are between two oppositely charged atoms. All ionic interactions are broken down into two subtypes, those mediated by a protein backbone, or side chains.

Water bridges are hydrogen-bonded protein-ligand interactions mediated by a water molecule. The hydrogen-bond geometry is slightly relaxed from the standard H-bond definition.

#### 4.9.4. Ligand Torsion Profile

The ligand torsions plot summarizes the conformational evolution of every rotatable bond (RB) in the ligand throughout the simulation trajectory (0.00 through 50.05 ns). The top panel shows the 2d schematic of a ligand with color-coded rotatable bonds. The bar plots summarize the data on the dial plots, by showing the probability density of the torsion. If torsional potential information is available, the plot also shows the potential of the rotatable bond (by summing the potential of the related torsions) [112].

Different ligand properties such as Ligand RMSD, Radius of Gyration, Intramolecular Hydrogen Bonds, Molecular Surface Area, Solvent Accessible Surface Area and Polar Surface Area were also measured. 

#### 4.9.5. Radius of Gyration (RoG)

Radius of Gyration used to measure compactness of Ligand-protein complex. RoG is usually plotted after MD simulation for a protein-ligand complex. This elucidates the stability of the complex in addition to the RMSD. An increase in RoG values implies a decrease in protein structure compactness, thereby suggesting increased flexibility and less stability. Radius of Gyration was calculated during 50 ns active binding of the Ligand–Protein complex [113].

### 4.10. Evaluation of Pharmacokinetics by Swiss ADME

ADMET (Absorption, Distribution, Metabolism, Excretion and Toxicity) parameters have been effective in the drug discovery phase of clinical trials for the pharmacokinetics-related non-performance of potential therapeutic drugs [114]. In this study, we utilized the *Swiss ADME* free web tool to evaluate the ADMET properties of several therapeutic agents. The *Swiss ADME* program evaluates pharmacokinetics properties, biological target prediction, molecular docking interactions, bio-isosteric design and molecular actions of chemical compounds. This tool requires SMILE or chemical structures as an input file for prediction. This user-friendly interface provides easy efficient input and interpretation through the free website, http://www.swissadme.ch (accessed on 10 July 2022). 

### 4.11. Physicochemical Descriptors and Lipophilicity Properties 

Physicochemical properties (descriptors) of therapeutic molecules are critically important in pharmacokinetics drug discovery [115]. Some of these descriptors include molecular refractivity (MR), topological polar surface area (TPSA), H-bond acceptors (HBA), H-bond donors (HBD), rotatable bonds, Fraction Csp^3^ (FCsp^3^), aromatic heavy atoms (atoms other than carbon) and heavy atoms (carbon atoms), molecular weight (MW), lipophilicity, solubility, GI absorption, skin permeability and P-glycoprotein efflux (P-gp) [116,117]. In our research we used the *Swiss ADME* program to calculate MR, TPSA and HBA, which are beneficial in predicting noteworthy therapeutic compounds. These properties are helpful regarding biological membrane penetration such as cellular absorption and drugs that cross the blood–brain barrier [118]. Physicochemical properties of a drug should be governed by Lipinski’s Rule of 5 (Ro5). This algorithm is important when selecting pharmacologically active constituents to increase the activity/selectivity and ensure drug likeness with optimal bioavailability [119]. Generally, Ro5 states that an orally active (bioactive) drug will have no more than one violation by the following criteria: ≤5 hydrogen (H) bond acceptors (nitrogen (N) or oxygen (O) atoms), ≤10 H bond donors (N or O atoms), molecular mass ≤ 500 g/mol and a partition coefficient Log P (cLog P) ≤ 5 [118].

Lipophilicity is the logarithm of solute concentration in octanol over a solute concentration in water (log P). Log Po/w represents the partition coefficient of *n*-octanol-water at a specific pH. Comparisons between commercial drugs and pre-clinical therapeutic compounds signify that log P values greater than 5 (per Ro5 criteria) may possess detrimental attributes such as rapid metabolic throughput, inadequate aqueous solubility, escalated plasma protein binding, accumulation of tissue, in vitro receptor promiscuity and in vivo toxicity. Conversely, if log P values are too low, then the potential drug could display inferior ADMET traits [120]. Log Po/w was calculated in our investigation by using five discrete predictive models: iLOGP [121], XLOGP3 [122], WLOGP [123], MLOGP [124] and Silicos-it [125]. It is important that predictor models should have varying levels of complexity to improve the prediction accuracy through a consensus model. Finally, the arithmetic mean of the values predicted by the five models can be interpreted in the form of the consensus log P value [115].

### 4.12. Water Solubility, Pharmacokinetics, Drug Likeness and Medicinal Chemistry

There are several characteristics and parameters required to develop an effective and safe drug. These properties include water solubility, pharmacokinetics, polarity, lipophilicity, molecule size, rotatability and drug likeness [126]. For example, water solubility is one of the major properties required for oral administration of a drug [118]. Two different methods, ESOL (Estimated Solubility) and Silicos-it, can predict a compound’s solubility in water. 

Pharmacokinetic properties also include the skin permeability coefficient (Kp), that is predicted by a multiple linear regression model [127]. Kp suggests a molecule’s role in transportation across the skin and epidermis, especially concerning hair follicles and sweat glands that form other pathways to the epidermis [118]. It has been proven that there is a direct linear correlation of Kp with molecular size and lipophilicity. Moreover, the more negative a log Kp value (cm/s) is, the less skin permeable is the affected molecule [128]. 

Information about molecular interactions with Cytochromes P450 (CYP450) is also important in drug design and development. CYP450s are a superfamily of isoenzymes that engage in the internal and extracellular component’s synthesis and metabolism. Isoenzyme inhibition can lead to adverse drug effects or the accumulation of drug metabolites. The CYP enzyme family and the permeability glycoprotein, P-gp, enhance the protection of tissues and organs against the entry of small molecules [129,130]. Interactivity of compounds with cytochrome P450 (CYP450) has been studied using isoforms such as CYP1A2, CYP2C19, CYP2C9, CYP2D6 and CYP3A4 [131,132]. Inhibition of any or all CYP isoforms leads assuredly to pharmacokinetic drug–drug interactions, leading in turn to adverse effects (AEs). It is critical to monitor AEs through the drug development process so that cytotoxicity and ultimately patient side effects may be kept to a minimum [133]. 

Absorption of drugs into a body’s metabolism is dependent on the administration. The most common route of administration is through the mouth. Most orally administered drugs are absorbed through the gastrointestinal (GI) tract. GI absorption is determined by the permeability of GI mucosa and the transit rate within the GI tract. It is also affected by the physiochemical state of the drug compound, intestinal physiology metabolic functions of the absorbing cells and the structure of the absorbing surface [130,134,135,136]. 

One critical factor in biological activity is the ability to orally administer a drug. Oral bioavailability of a compound is the rate and extent that an active pharmaceutical therapeutic is readily absorbed and then available to the circulation of the body [137]. Based on physicochemical properties of clinical trial phase II drugs, oral bioavailability can be predicted by Ro5 [138]. 

Finally, in analyzing the medicinal chemistry of a compound, two properties, PAINS (pan-assay interference compounds) and synthetic accessibility (SA) can be studied. To identify problematic fragments in our research, PAINS (frequent hitters or promiscuous compounds) were evaluated. SA was utilized to select the most effective potential virtual pharmaceutical compounds that could be synthesized in order subsequently to be evaluated by biological experimentation. SA is therefore a major selective criterion in the process of the development of suitable drug compounds [73]. 

## 5. Conclusions

COVID-19 is caused by the infection of the novel coronavirus, SARS-CoV-2 and may continue to present a severe global threat for some time to come. In spite of earlier major outbursts of CoV infections of SARS and MERS, there is a lack of effective drug interventions to combat the novel disease. It is very difficult to produce productive therapeutic treatments and vaccines against future strains of SARS-CoV-2 variants and other highly infectious pathogenic viruses. Tools and therapeutics to combat outbreaks are essential to decrease the ruinous impact on human populations and global economies. Experimental evaluation, clinical drug designing and development of large sets of therapeutic compounds is expensive and a time-consuming procedure—often taking 10–15 years from pre-clinical studies to government approved drugs. 

It is known that several species of algae are good reserves of many primary and secondary metabolites such as sulphated polysaccharides, phenolic compounds, proteins, fatty acids, amino acids, lectins and pigments that possess strong antiviral properties. These phyco-molecules are predicted to be drug-like, as well as nontoxic, noncarcinogenic and nonmutagenic. In our hands, a variety of these bioactive molecules were found to be effective therapeutic compounds against some of the selected target proteins by in silico molecular docking techniques. The outcomes of our evaluations will be additionally helpful for future in vitro and in vivo examinations to provide drug designs and pharmacological evaluations. According to our findings, 3,7,11,15-Tetramethyl-2-hexadecen-1-ol is the natural phyco-compound from green alga, *U. fasciata* that should be studied and characterized more thoroughly as novel therapeutic compounds that may prevent or treat SARS-CoV-2 infections after in vivo experimental evaluation. The identification of target binding sites and inhibitory mechanism will ultimately advance the understanding of antiviral actions in humans and lead to viable disease treatment.

## Figures and Tables

**Figure 1 marinedrugs-20-00586-f001:**
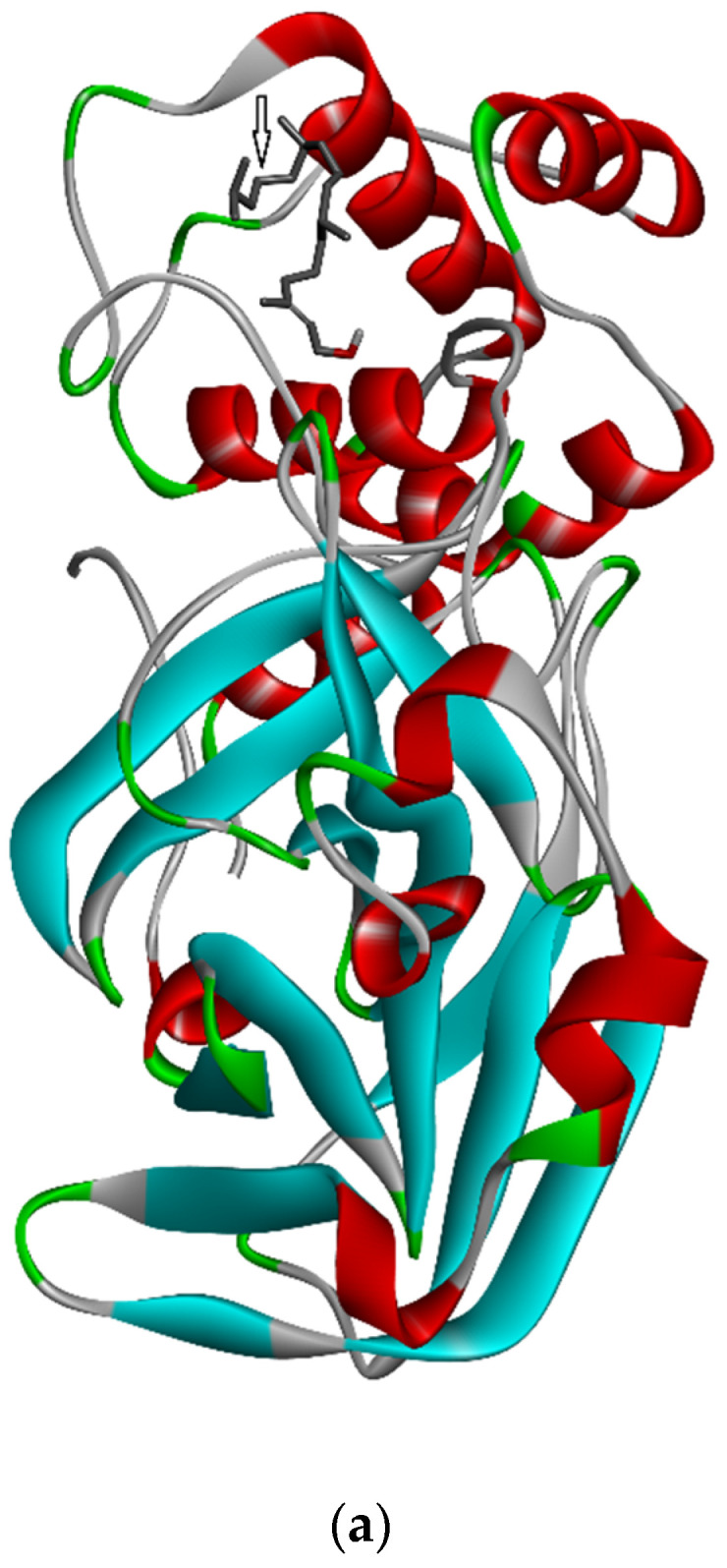
Molecular interactions between ligand, 3,7,11,15-Tetramethyl-2-hexadecen-1-ol and SARS-CoV-2 target protein, 6Y84: (**a**) 3D interaction of ligand and protein, Arrow indicates structures of ligand and its interaction with target protein; different colored bands show different amino acid chains of target proteins; (**b**) non-bond interaction between receptor and ligand (with interacting atoms and pocket atoms), Arrow indicates ligand molecule grey in color, other labels and distance values show its interaction with target protein; (**c**) 2D representation of 3D ligand binding site; (**d**) Ramachandran plot (Green: Inside; Pink: Overlap; Turquoise blue: Hardsphere); (**e**) Receptor surface, 1: Aromatic, 2: H-Bonds, 3: Interpolated charge, 4: Hydrophobicity, 5: Ionizability, 6: SAS.

**Figure 2 marinedrugs-20-00586-f002:**
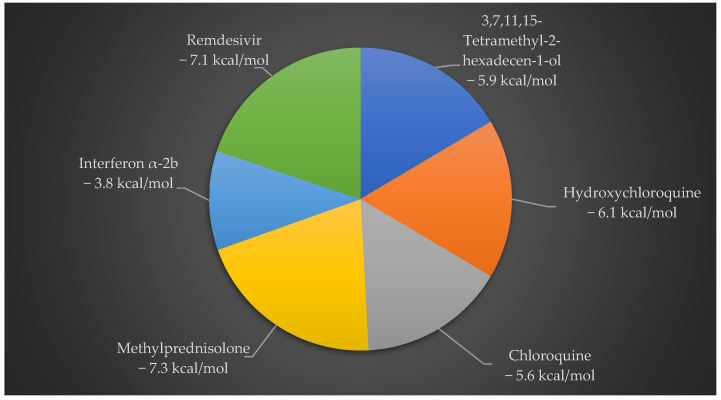
Docking interaction analysis of binding energies between 3,7,11,15-Tetramethyl-2-hexadecen-1-ol and 5 standard antiviral drugs on SARS-CoV-2 6Y84 protein. Of the 5 drug compounds, 3,7,11,15-Tetramethyl-2-hexadecen-1-ol exhibits an optimal binding pose with 6Y84 over the other 2 standard antiviral therapeutic drugs.

**Figure 3 marinedrugs-20-00586-f003:**
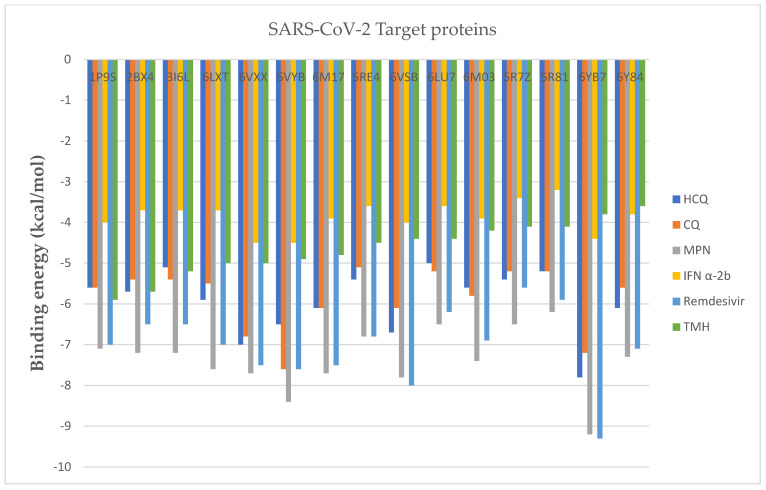
Binding energies of 5 standard antiviral drugs and 1 phyco-compound with 15 SARS-CoV-2 target proteins.

**Figure 4 marinedrugs-20-00586-f004:**
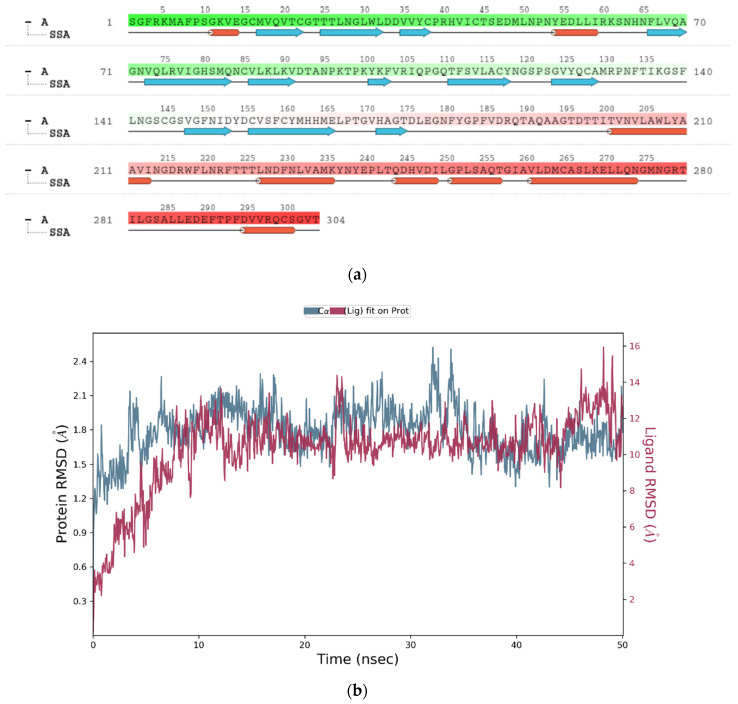
(**a**) Graphical illustration for protein A chain. 304 resdiues in Protein chain A of selected target protein, 4646 atoms, 2348 heavy atoms, charge: −3, different color shows pattern of SSA in 304 amino acid residues. (**b**) The RMSD evolution of a protein (left Y-axis) and ligand (left X-axis). (**c**) Protein fluctuation study during simulation through RMSF. (**d**) SSE distribution by residue index in protein structure. (Protein secondary structure, % Helix: 16.08; % Strand: 25.01; % Total SSE: 41.09). Saffron color indicates alpha-helices; Blue color indicates beta-strands. (**e**) SSE composition for each frame. (**f**) Residue and its SSE assignment over time. Saffron color: alpha-helices; Blue color: beta-strands. (**g**) Ligand RMSF and its interaction with the 6Y84 protein. (**h**) Protein–ligand interactions (or ‘contacts’) throughout the simulation. (**i**) Total number of protein–ligand contacts. (**j**) Residues interactions with the ligand. (**k**) A schematic interactions of ligand atom with the protein residues. (**l**) Ligand Torsion Profile. The top panel represents 2D schematic of a ligand with color-coded rotatable bonds. The bottom panel represents each rotatable bond torsion accompanied by a dial plot and bar plots of the same color. (**m**) Ligand properties: ligand RMSD, rGyr, intraHB, MolSA, SASA and PSA. (**n**) Radius of Gyration calculated for 50 ns in MD simulation.

**Figure 5 marinedrugs-20-00586-f005:**
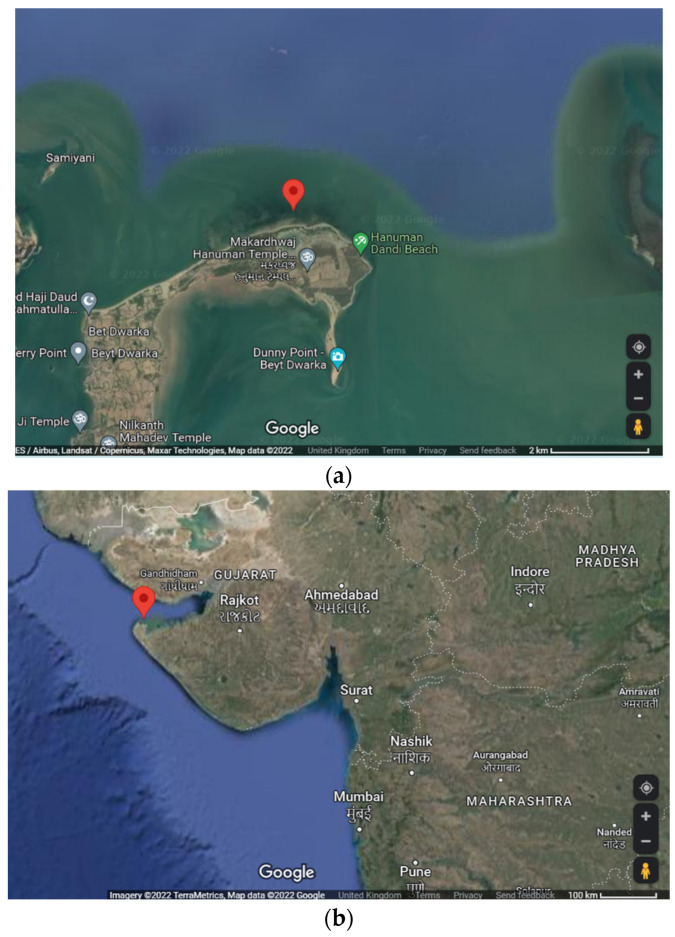
Geographical views of sample site: (**a**) location of sampling site at 22°28′47.9″ N, 69°08′05.0″ E, (**b**) location of sampling site in respect to state of Gujarat, India.

**Figure 6 marinedrugs-20-00586-f006:**
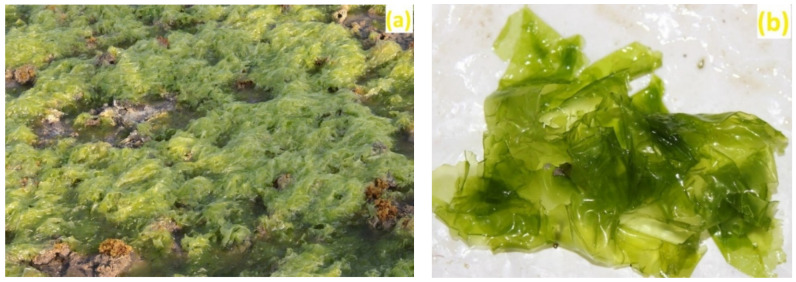
*Ulva fasciata* (Chlorophyta): (**a**) onshore view of the macroalga in sea water, (**b**) isolated sample of *U. fasciata*.

**Table 1 marinedrugs-20-00586-t001:** PubChem^®^ Phyco-compounds Isolated from *U. fasciata*.

Compound	PubChem^®^ ID	Mol. Formula	Mol. Weight	CAS ID	SMILE Structure
Azelaic acid	2266	C_9_H_16_O_4_	188.22 g/mol	123-99-9 27825-99-6 26776-28-3	C(CCCC(=O)O)CCCC(=O)O
NI	935	Ni	58.693 g/mol	7440-02-0 14903-34-5	[Ni]
*n*-Pentadecanoic acid	13849	C1_5_H_30_O_2_	242.4 g/mol	1002-84-2	CCCCCCCCCCCCCCC(=O)O
Hexahydro farnesyl acetone	10408	C_18_H_36_O	268.5 g/mol	502-69-2 16825-16-4	CC(C)CCCC(C)CCCC(C)CCCC(=O)C
Palmitic acid	985	C_16_H_32_O_2_	256.42 g/mol	57-10-3 67701-02-4	CCCCCCCCCCCCCCCC(=O)O
Palmitic acid ethyl ester	12366	C_18_H_36_O_2_	284.5 g/mol	628-97-7	CCCCCCCCCCCCCCCC(=O)OCC
Trichloromethyl-oxirane	18321	C_3_H_3_Cl_3_O	161.41 g/mol	3083-23-6	C1C(O1)C(Cl)(Cl)Cl
3,3,5-Trimethylhexahydro-azepine	118239	C_9_H_19_N	141.25 g/mol	35466-89-8	CC1CCNCC(C1)(C)C
2-Butyl-1-octanol	19800	C_12_H_26_O	186.33 g/mol	3913-02-8	CCCCCCC(CCCC)CO
3,7,11,15-Tetramethyl-2-hexadecen-1-ol	5366244	C_20_H_40_O	296.5 g/mol	7541-49-3	CC(C)CCCC(C)CCCC(C)CCCC(=CCO)C
Phytol	5280435	C_20_H_40_O	296.5 g/mol	150-86-7	CC(C)CCCC(C)CCCC(C)CCCC(=CCO)C
Docosanoic acid, methylester	13584	C_23_H_46_O_2_	354.6 g/mol	929-77-1	CCCCCCCCCCCCCCCCCCCCCC(=O)OC

**Table 2 marinedrugs-20-00586-t002:** Toxicity predictions by VEGA QSAR for 6 types of mutagenicity/carcinogenicity/toxicity analyses on select therapeutic compounds.

Therapeutic Compound	Mutagenicity (Ames Test) CONSENSUS Model 1.0.3	Mutagenicity (Ames Test) Model (CAESAR) 2.1.13	Carcinogenicity Model (CAESAR) 2.1.9	Carcinogenicity Oral Classification Model (IRFMN) 1.0.0	Developmental Toxicity Model (CAESAR) 2.1.7	Developmental/ Reproductive Toxicity Library (PG) 1.1.0
Azelaic acid	NM (0.9)	NM (0.922)	NC (0.748)	NC (0.851)	NT (0.816)	NT (0.883)
NI	NM (0.2)	NM (−)	Not calculated	Not calculated	NT (0.38)	NT (0.426)
*n*-Pentadecanoic acid	NM (0.9)	NM (0.969)	NC (0.575)	NC (0.757)	NT (0.848)	NT (0.887)
Hexahydro-farnesyl acetone	NM (0.675)	NM (0.84)	NC (0.502)	NC (0.744)	NT (0.767)	NT (0.794)
Palmitic acid	NM (1)	NM (0.965)	NC (0.575)	NC (0.753)	NT (0.846)	NT (0.874)
Palmitic acid ethyl ester	NM (0.825)	NM (0.914)	NC (0.77)	NC (0.802)	NT (0.847)	NT (0.851)
Trichloromethyl-oxirane	NM (1)	NM (1)	C (0.826)	C (0.815)	T (0.628)	T (0.824)
3,3,5-Trimethylhexa-hydroazepine	NM (0.825)	NM (0.862)	NC (0.59)	C (0.797)	NT (0.718)	NT (0.871)
2-Butyl-1-octanol	NM (0.825)	NM (0.925)	NC (0.945)	C (0.776)	T (0.82)	T (0.882)
3,7,11,15-Tetramethyl-2-hexadecen-1-ol	NM (0.825)	NM (0.814)	C (0.655)	NC (0.691)	NT (0.807)	NT (0.799)
Phytol	NM (0.825)	NM (0.814)	C (0.655)	NC (0.691)	NT (0.807)	NT (0.799)
Docosanoic acid, methyl ester	NM (0.75)	NM (0.893)	NC (0.87)	NC (0.795)	T (0.808)	NT (0.813)

Non-mutagenic: NM; Non-carcinogenic: NC; Non-toxic: NT; Carcinogen: C; Toxic: T.

**Table 3 marinedrugs-20-00586-t003:** Outcomes of PASS predictions for 5 therapeutic compounds against select viruses (*Pa = probable biological activity of drug; +Pi = probably biological inactivity of drug).

PubChem Name	PubChem ID	*Pa	+Pi	Viruses
Azelaic acid	2266	0.670	0.008	Picornavirus
		0.641	0.013	Poxvirus
		0.596	0.007	Rhinovirus
		0.524	0.019	Influenza
		0.508	0.005	Adenovirus
Pentadecanoic acid	13849	0.671	0.008	Picornavirus
		0.611	0.005	Rhinovirus
		0.608	0.014	Poxvirus
		0.565	0.016	Influenza
		0.519	0.005	Adenovirus
		0.502	0.003	Cytomegalovirus
Palmitic acid	985	0.671	0.008	Picornavirus
		0.611	0.005	Rhinovirus
		0.608	0.014	Poxvirus
		0.565	0.016	Influenza
		0.519	0.005	Adenovirus
		0.502	0.003	Cytomegalovirus
Ethyl palmitate	12366	0.695	0.006	Picornavirus
		0.691	0.003	Rhinovirus
		0.556	0.004	Adenovirus
		0.523	0.002	Cytomegalovirus
		0.508	0.021	Influenza
Hexahydro-farnesyl acetone	10408	0.464	0.040	Rhinovirus
		0.449	0.076	Picornavirus
		0.383	0.036	Adenovirus
		0.368	0.057	Influenza
		0.303	0.027	Cytomegalovirus
		0.270	0.078	Poxvirus

Toxicology data for the 5 potential drugs show that Pa is between 0.5–0.7 for 6 known viruses. The compounds may be considered structurally novel therapeutics against SARS-CoV-2.

**Table 4 marinedrugs-20-00586-t004:** Ligand-protein binding interaction predictions of SARVS-CoV-2 target proteins and therapeutic compounds by AutoDock Vina (binding energies in kcal/mol).

SARS-CoV-2 Target Protein	985	2266	10408	13584	13849	18321	19800	118239	12366	5280435	5366244
PDB ID:											
1P9S	−4.9	−4.7	−5.5	−4.7	−4.6	−3.6	−4.8	−5.3	−4.7	−5.3	−5.7
2BX4	−3.5	−3.5	−4.3	−4.0	−3.9	−2.9	−3.8	−4.3	−3.3	−4.7	−4.5
3I6L	−3.4	−5.1	−4.2	−3.5	−4.8	−3.5	−3.9	−4.6	−3.1	−4.1	−3.6
6LXT	−2.8	−3.7	−3.3	−2.6	−3.3	−4.4	−3.4	−4.7	−2.5	−2.7	−4.1
6VXX	−4.7	−5.5	−5.3	−5.3	−4.8	−3.5	−5.4	−5.2	−4.4	−5.6	−5.0
6VYB	−4.7	−4.5	−5.2	−5.0	−4.5	−3.4	−4.3	−5.0	−4.3	−5.4	−4.8
6M17	−4.2	−4.3	−4.8	−4.2	−4.5	−3.2	−4.1	−5.2	−4.0	−4.6	−5.0
5RE4	−3.1	−3.7	−3.7	−3.7	−4.5	−3.4	−4.0	−4.6	−3.1	−4.3	−3.8
6VSB	−4.6	−4.3	−4.5	−4.1	−4.9	−3.3	−4.3	−4.5	−4.2	−4.8	−5.2
6LU7	−3.9	−4.1	−4.1	−3.8	−3.9	−3.4	−3.9	−4.9	−3.7	−3.9	−4.4
6M03	−3.9	−4.3	−4.1	−4.8	−4.7	−3.5	−4.3	−4.8	−3.3	−5.2	−4.4
5R7Z	−3.2	−4.5	−3.8	−3.5	−3.5	−3.5	−4.0	−4.7	−3.1	−3.8	−4.1
5R81	−3.6	−4.1	−4.4	−3.2	−4.0	−3.4	−3.8	−4.7	−3.3	−4.9	−4.2
6YB7	−5.5	−4.8	−4.8	−4.4	−4.7	−3.5	−4.6	−5.3	−5.1	−5.1	−4.9
6Y84	−4.8	−4.8	−4.9	−4.9	−4.4	−3.3	−4.6	−5.0	−4.4	−5.7	−5.9

Yellow highlights show the therapeutic compound that possess the lowest binding energy value to target protein: 3,7,11,15-Tetramethyl-2-hexadecen-1-ol with 6Y84 (−5.9 kcal/mol).

**Table 5 marinedrugs-20-00586-t005:** Physicochemical and lipophilic properties of 3,7,11,15-Tetramethyl-2-hexadecen-1-ol by *Swiss ADME*.

-	3,7,11,15-Tetramethyl-2-hexadecen-1-ol
Physicochemical Properties	
Heavy atoms	21
Aromatic heavy atoms	0
Fraction Csp^3^	0.90
Rotatable bonds	13
H-bond acceptors	1
H-bond donors	1
MR	98.94
TPSA	20.23 Å^2^
Lipophilic Properties	
iLOGP	4.71
XLOGP3	8.19
WLOGP	6.36
MLOGP	5.25
Silicos-it Log P	6.57
Consensus Log P	6.22

**Table 6 marinedrugs-20-00586-t006:** Water solubility and pharmacokinetic properties of 3,7,11,15-Tetramethyl-2-hexadecen-1-ol by *Swiss ADME*.

	3,7,11,15-Tetramethyl-2-hexadecen-1-ol
Water solubility	
ESOL class	Moderately soluble −5.98
Silicos-it class	Moderately soluble −5.51
Pharmaco-kinetics	
GI absorption	Low
P-gp substrate	Yes
CYP2C19 inhibitor	No (0.96)
CYP2C9 inhibitor	No (0.827)
CYP2D6 inhibitor	No (0.842)
CYP3A4 inhibitor	No (0.99)
Skin permeability log Kp (cm/s)	−2.29 cm/s
Drug Likeness	
Lipinski Rule of 5	1
Bioavailability score	0.55
Medicinal Chemistry	
PAINS alerts	0
Synthetic accessibility	4.30

In CYPs inhibition, value represents strength of probability for inactive class. Strength of prediction: Dark green: Strong.

**Table 7 marinedrugs-20-00586-t007:** SARS-CoV-2 target proteins from PDB for docking interaction analysis.

Protein ID	Protein Structure and Function Characteristics	References
1P9S	Main proteinase (3CL^pro^) structure	[90]
2BX4	Crystal structure of main proteinase (P21212)	[91]
3I6L	Epitope N1 derived from SARS-CoV N protein complexed with HLA-A*2402	[44]
6LXT	Post-fusion core of 2019-nCoV S2 subunit	[92]
6VXX	SARS-CoV-2 spike glycoprotein (closed state)	[93]
6VYB	SARS-CoV-2 spike ectodomain (open state)	[93]
6M17	RBD/ACE2-B0AT1 complex	[94]
5RE4	SARS-CoV-2 main protease in complex with Z1129283193	[95]
6VSB	Prefusion 2019-nCoV spike glycoprotein with a single receptor-binding domain up	[96]
6LU7	Crystal structure main protease in complex with an inhibitor N3	[45]
6M03	Crystal structure of main protease in apo form	[97]
5R7Z	SARS-CoV-2 main protease in complex with Z1220452176	[96]
5R81	Crystal structure of main protease in complex with Z1367324110	[96]
6YB7	SARS-CoV-2 main protease with unliganded active site	[98]
6Y84	SARS-CoV-2 main protease with unliganded active site	[98]

**Table 8 marinedrugs-20-00586-t008:** PubChem^®^ standard drugs with chemical characteristics.

Drugs	PubChem^®^ ID	Molecular Formula	Molecular Weight	CAS ID	SMILE
Hydroxychloroquine	3652	C_18_H_26_ClN_3_O	335.9	118-42-3	CCN(CCCC(C)NC1=C2C=CC(=CC2=NC=3C1)Cl)CCO
Chloroquine	2719	C_18_H_26_ClN_3_	319.9	54-05-7	CCN(CC)CCCC(C)NC1=C2C=CC(=CC2=NC=C1)Cl
Methyl-prednisolone	6741	C_22_H_30_O_5_	374.5	83-43-2	CC1CC2C3CCC(C3(CC(C2C4(C1=CC(=O)C=C4)C)O)C)(C(=O)CO)O
Interferon α-2b	71306834	C_16_H_17_Cl_3_I_2_N_3_NaO_5_S	746.5	98530-12-2	CCCN(CCOC1=C(C=C(C=C1Cl)Cl)Cl)C(=O)N2C=CN=C2.C(S(=O)(=O)[O-])(I)I.[Na+]
Remdesivir	121304016	C_27_H_35_N_6_O_8_P	602.6	1809249-37-3	CCC(CC)COC(=O)C(C)NP(=O)(OCC1C(C(C(O1)(C#N)C2=CC=C3N2N=CN=C3N)O)O)OC4=CC=CC=C4

## Data Availability

Not applicable.

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
