# Peer review of "Marine Alga Ulva fasciata-Derived Molecules for the Potential Treatment of SARS-CoV-2: An In Silico Approach"

_marinedrugs, 2022, doi:10.3390/md20090586_

Round 1

Reviewer 1 Report

Please answer thoroughly to my comments.

Author Response

Reviewer 1

 Minor comments:

  1. The introduction is too long and should be shortened.

Response: Removed unnecessary content that showed a very basic information for CoV.

  1. Table 3 is filled with empty spaces where chemical formulas probably should be placed instead.

Response: Corrected, for each compound 4-5 antiviral activities predicted.

  1. The resolution of Figure 3 should be increased.

Response: Improved

  1. The study lacks molecular dynamics analysis of the most stable protein-ligand structures: the molecular docking method is not accurate enough to measure binding energies.

Response: I ran all the compounds for molecular docking interaction study (binding energy) by AutoDock Vina instead of iGemDock software. Because another reviewer said to do Mol. Docking for all the selected compounds.

From the obtained data, we selected top first compound (with optimum result of binding energy) and did MD simulation study by Desmond of Schrodinger’s suit. We also studied other parameters (RMSD, RMSF, Radius of Gyration, Ligand Torsion Profile, residues interaction, etc.) and plotted its graphs in MS with yellow highlight.

  1. Section 4.4 can be deleted since it is trivial.

Response: Deleted and made all the changes in manners. Also incorporated some latest references and references for newly added data of MD simulation results and methods.

Reviewer 2 Report

This publication concerns the search for new potential drugs against SARSCov2 from compounds isolated the seaweed Ulva fasciata.  It is one of the numerous publications using the in silico docking approach in the search for new  COVID19 drugs. Unfortunately, this approach, without confirmation by in-vitro methods due to the low reliability of the docking methods, is only speculative and, in my opinion, it requires an approach using molecular dynamics simulation methods in order to refine the obtained results.

At the same time, in the case of docking, it is crucial to assess the binding site, in order to achieve a therapeutic effect it is necessary that the interaction be specific with the active (regulatory) site of the protein. In the case of this study, it seems that the active site in the target protein was not defined, which could lead to non-specific surface interaction (which can be seen in Figure 1).
It is also worth noting that in some of the analyzed protein structures, inhibitors create a covalent interaction with the protein (6LU7), which should be commented on in the context of the results obtained.

It is a pity that the authors have eliminated some compounds from further stages of work at the stage of toxicological analysis. Docking all 11 compounds would allow assessing the therapeutic potential of all compounds, not just the 5 selected
Even more , the choice of PA compound seems a bit surprising as it is a standard component of vegetable and animal fats.

In my opinion, the publication should be improved, in the case of docking, the protein active site should be clearly defined, the docking results in terms of bioactive conformation should compared with model compounds (not only in terms of docking scores), and the interaction energies and the stability of the complex should be refined using the MD simulation.

Author Response

Reviewer 2

Comment: In my opinion, it requires an approach using molecular dynamics simulation methods in order to refine the obtained results.

Response: I did MD simulation of top first compound (With highest result of Binding energy in Mol. Docking study) by Desmond from Schrodinger’s suit.

I have also analyzed different parameters in DM simulation such as RMSD, RMSF, Ligand torsion profile, Residues interaction, Ligand properties, Radius of gyration as well as other interaction properties.

Comment: It is a pity that the authors have eliminated some compounds from further stages of work at the stage of toxicological analysis. Docking all 11 compounds would allow assessing the therapeutic potential of all compounds, not just the 5 selected
Response: I did molecular docking study of all the compounds by AutoDock Vina instead of iGemDock due to high performance of its action. we selected the optimal compound and proceeded for MD simulation study.

We also added docking study of standard drugs by AutoDock Vina and compared its result with ligands output. Meanwhile, added new and related references also incorporated 3D and 2D interaction of Ligand-Protein Complexes. Please find yellow highlighted text for improved MS data.
